## Protocol for community-driven selection of strategies to implement evidence-based practices to reduce opioid overdoses in the HEALing Communities Study: a trial to evaluate a community-engaged intervention in Kentucky, Massachusetts, New York and Ohio

April M Young [1], Jennifer L Brown,[2] Timothy Hunt,[3] Linda S Sprague Martinez,[4] Redonna Chandler,[5] Emmanuel Oga,[6] T John Winhusen,[7,8] Trevor Baker,[9] Tracy Battaglia,[10] Rachel Bowers-Sword,[9] Amy Button,[11] Amanda Fallin-Bennett,[12] Laura Fanucchi,[13] Patricia Freeman,[14] LaShawn M Glasgow,[15] Jennifer Gulley,[16] Charles Kendell,[17] Michelle Lofwall,[13] Michael S Lyons,[18] Maria Quinn,[19] Bruce David Rapkin,[20] Hilary L Surratt,[13] Sharon L Walsh[13]

For numbered affiliations see end of article.

**Correspondence to**
Dr April M Young;
april.young@uky.edu

## ABSTRACT

**Introduction** Opioid-involved overdose deaths continue to surge in many communities, despite numerous evidence-based practices (EBPs) that exist to prevent them. The HEALing Communities Study (HCS) was launched to develop and test an intervention (ie, Communities That HEAL (CTH)) that supports communities in expanding uptake of EBPs to reduce opioid-involved overdose deaths. This paper describes a protocol for a process foundational to the CTH intervention through which community coalitions select strategies to implement EBPs locally.

**Methods and analysis** The CTH is being implemented in 67 communities (randomised to receive the intervention) in four states in partnership with coalitions (one per community). Coalitions must select at least five strategies, including one to implement each of the following EBPs: (a) overdose education and naloxone distribution; expanded (b) access to medications for opioid use disorder (MOUD), (c) linkage to MOUD, (d) retention in MOUD and (e) safer opioid prescribing/dispensing. Facilitated by decision aid tools, the community action planning process includes (1) data-driven goal setting, (2) discussion and prioritisation of EBP strategies, (3) selection of EBP strategies and (4) identification of next steps. Following review of epidemiologic data and information on existing local services, coalitions set goals and discuss, score and/or rank EBP strategies based on feasibility, appropriateness within the community context and potential impact on reducing opioid-involved overdose deaths with a focus on three key sectors (healthcare, behavioural health and criminal justice) and high-risk/vulnerable populations. Coalitions then select EBP strategies through consensus or majority vote and, subsequently, suggest or choose agencies with which to partner for implementation.

## STRENGTHS AND LIMITATIONS OF THIS STUDY

⇒ A strength of the Communities That HEAL (CTH) action planning protocol is that it balances scientific rigour with community priorities by ensuring all communities follow a common core of structured activities while also allowing variation across states and communities to best suit the local context.

⇒ The methods used in the CTH action planning process elevate the voices of community members, whose insights are critical to implementation success and whose involvement conveys to local stakeholders that evidence-based practice (EBP) implementation is a priority of the local community.

⇒ The action planning protocol requires careful facilitation by coalition members and community staff to ensure that conflicts of interest are managed and power dynamics do not result in some coalition members having more influence than others in the decision-making process.

⇒ The phased and iterative action planning protocol is time-intensive requiring significant commitment from involved community members, faculty and staff and may slow implementation of EBPs.

⇒ While necessitated by the constraints of the study timeline, the research teams' intensive involvement in facilitating the action planning protocol could be a limitation as it may stifle participation by coalition members who approach the process with deference and thereby limit the extent to which it is community driven.

**Ethics and dissemination** The HCS protocol was approved by a central Institutional Review Board (Advarra). Results of the action planning process will be disseminated in academic conferences and peer-reviewed journals, online and print media, and in meetings with community stakeholders.

**Trial registration number** NCT04111939.

## INTRODUCTION

Opioid-involved overdose deaths remain a public health crisis in many communities worldwide[1 2] although numerous evidence-based practices (EBPs) exist to prevent them. In the USA alone, opioid-involved overdose claimed over 490 000 lives between 1999 and 2019.[3] Despite national efforts to change the trajectory of overdose deaths, provisional data indicate over 94 000 total overdose deaths occurred in the 12 months ending in January 2021, making this the highest number of overdose deaths ever recorded in a 12-month period.[4]

Research has demonstrated that medication for opioid use disorder (MOUD), including methadone, buprenorphine and extended-release naltrexone;[5] prescription drug monitoring programmes and safer prescribing guidelines for prescription opioids[6]; naloxone, a medication that can reverse an opioid overdose[7] and behavioural/psychosocial interventions like recovery support services[5] can prevent deaths and help people enter and stay in remission and recovery from opioid use disorder (OUD). Unfortunately, these EBPs remain underutilised in many communities leaving significant unmet need for these critically important services.[8–15]

The HEALing Communities Study (HCS), supported by the National Institutes of Health and the Substance Abuse and Mental Health Services Administration, launched in 2019 to develop and test an approach to support communities in expanding uptake and availability of EBPs to address opioid-involved overdose deaths and OUD. HCS is a multisite, parallel-group, cluster randomised waitlist-controlled trial involving 67 communities (34 in active intervention and 33 in waitlist control) in Kentucky (KY), Massachusetts (MA), New York (NY) and Ohio (OH).[16] The study is designed to test the Communities That HEAL (CTH) intervention, which contains three components: a community-engaged process of data-driven decision-making to select EBP strategies for local implementation and monitoring[17]; a menu of EBPs and technical assistance guides (Opioid-Overdose Reduction Continuum of Care Approach (ORCCA)),[18] and a set of communication campaigns intended to address stigma and increase knowledge of and demand for EBPs.[19] The CTH intervention is being implemented in the active intervention communities from January 2020 to June 2022 and will be implemented in waitlist control communities from July 2022 to December 2023.

The CTH intervention is grounded in the recognition that communities affected by the opioid overdose crisis possess the best understanding of local strengths, gaps, barriers and facilitators related to local EBP implementation and uptake.[17] Thus, the CTH involves partnerships between academic researchers and community members through community coalitions. Coalitions are comprised of representatives from diverse agencies and institutions as well as individuals focused on tackling complex community issues through collective planning.[20] The CTH intervention employs a phased coalition planning process that was developed based on and expands the Communities that Care model,[17] which has supported communities in the selection of evidence-based prevention efforts in previous research.[21–23] Coalition planning models are effective because they allow communities to leverage resources and expertise from across multiple sectors, which can facilitate creative problemsolving and innovation, reduce duplication of efforts, and address service gaps.[24 25] The literature points to coalition planning as an effective strategy for implementing EBPs.[26–28]

The purpose of this paper is to describe the protocol for coalition action planning through which EBP strategies are selected to reduce opioid overdose deaths in their communities.[16] The protocol described in this manuscript is relevant beyond the field of substance use, as community engagement and community-driven decision-making are recommended for improving responses to a variety of health challenges,[29–32] including, most recently, SARS-CoV-2 vaccination[33] and human Monkeypox.[34]

## METHODS AND ANALYSIS
### Participant involvement

As described elsewhere,[17] community engagement is central to the CTH intervention. The CTH intervention places communities, specifically community coalitions, at the centre of the decision-making process in creating an action plan for implementing EBP strategies locally. The coalitions are comprised of key stakeholders in healthcare, behavioural health, criminal justice, public health, government, business, housing and other sectors, and, in many cases, include people with lived experience with or impacted by OUD. In some communities, the coalitions that are presently engaged in the CTH existed prior to the HCS, while others were formed specifically to guide the CTH intervention. In addition to driving the CTH intervention, these coalitions facilitate dissemination of information about the study and about EBPs to other local stakeholders. Coalitions, local agency partners and partnerships formed between academic institutions and local stakeholders will also be key to sustainability of the CTH process and EBPs. Of note, coalitions were not involved in setting the research question, designing the intervention protocol or menu of EBPs or establishing outcome measures. Of note, each of the four participating states has a Community Advisory Board, with multisectoral representation from across the communities that advises the research teams on study implementation.

### Overview of CTH phases

The CTH intervention includes seven phases, described in detail elsewhere,[17] that guide coalitions through a

systematic partnership process. The intervention began 23 October 2019. The anticipated trial completion date is 31 December 2023. Briefly, the process begins with establishment or identification of a coalition and an environmental scan, or 'landscape analysis', of local resources and gaps (phase 0). During phase 1, coalitions develop a shared charter and identify key community 'champions' to support EBPs, data-driven decision-making and communication campaigns that have been described elsewhere.[19] In phase 2, coalitions learn about data-driven decision-making and the ORCCA menus containing EBP strategies related to overdose education and naloxone distribution (OEND, menu 1), MOUD (menu 2) and prescription opioid safety (menu 3). The ORCCA menus have been described in detail elsewhere.[18] Following review of data describing the local epidemiology of OUD and overdose, simulation modelling data and/or existing service resources and gaps identified by the landscape analysis (phase 3), coalitions develop community action plans (one per coalition) by selecting strategies to implement EBPs that are in the ORCCA menu (phase 4). After creation of the action plan, coalition-selected EBP strategies are implemented and monitored (phase 5) and sustainability plans are created (phase 6). While these processes are ongoing, a health communication campaign is delivered at the community level to reduce stigma and increase engagement in OEND and MOUD.[19] This paper describes the protocol for coalition planning efforts in phase 4 to select EBP strategies to achieve the study's primary outcome, reduction in number of opioid overdose deaths. The primary and secondary outcomes of the study and the analysis of the trial data are described in detail elsewhere.[35 36]

### Strategies included in the Community Action Planning Process

Coalitions are provided with guidelines to structure strategy selection in the community action planning process (figure 1). Coalitions are required to select a minimum of five strategies, including at least one to implement each required EBP from the ORCCA: (a) expansion of proactive offering of OEND to at risk individuals and members of their social networks, (b) expansion of buprenorphine and/or methadone access, (c) improved linkage to MOUD, (d) increased retention in MOUD and (e) promotion of safer opioid prescribing/dispensing. The EBP strategies are required to include at least one to be implemented in each of the following three sectors: healthcare, behavioural health and criminal justice.[18] Coalitions are encouraged to consider EBP strategies that would maximise impact in key populations most vulnerable to experiencing an opioid-involved overdose, including those who had a prior opioid overdose[8 37–40]; have reduced opioid tolerance (eg, completing medically supervised or socially managed withdrawal or release from institutional settings such as jail, residential treatment, hospital)[40–44]; use other substances (eg, alcohol, benzodiazepines, cocaine, and amphetamine like substances)[40 45–49]; have concomitant major mental[50–53] or

medical illness[40 54–57] and/or inject drugs.[46 58] Coalitions are also asked to prioritise venues through which these key populations are most likely to be reached. Priority venues for each of the three EBPs (ie, OEND, MOUD and safer opioid prescribing and dispensing) are listed in figure 1 and described in detail elsewhere.[18]

Inherently, the CTH allowed for communities to independently implement EBP strategies at their own pace, including expediting strategies in times of emergency. In June 2020, the HCS approved 'fast tracking' certain HCS-supported activities related to OEND in response to the coalitions' concerns about the substantial uptick of opioid overdose observed in the initial months of the SARS-CoV-2 pandemic.[59–66] The protocol allowed HCS-supported naloxone distribution to begin in high-risk settings prior to the completion of action planning but preserved the community-driven nature of the decision-making process. Coalitions were asked whether they wanted the HCS teams to provide technical assistance and/or resources to community venues with greatest reach to individuals at risk for overdose (ie, jails, syringe service programmes) to expand OEND. On coalition approval, HCS teams and/or community implementation teams began outreach to high-risk venues to expand OEND. Thus, the fast-tracked OEND activities were ongoing during the phase 4 action planning process. During the action planning process, coalitions were asked to reflect on the successes and challenges of the fast-tracked strategies and asked to decide whether they wanted to continue or expand OEND activities that had been fast-tracked as part of the action planning process.

### Strategy selection and action planning process

The core components of the cross-state protocol for action planning and strategy selection are described in figure 2. Action planning in each state involves (1) data-driven community goal setting, (2) discussion and prioritisation of EBP strategies, (3) selection of EBP strategies to include in the action plan and (4) identification of next steps for implementation of selected EBP strategies.

Coalitions' movement through the action planning process is facilitated by a combination of goal-setting tools, data visualisations and decision aids. Examples of tools developed by each state for the action planning process are provided in online supplemental appendix. The communities currently receiving the CTH intervention differ in geographic unit (ie, counties, townships, cities), community-readiness for EBP strategy implementation, existing resources and gaps, coalition history (eg, whether coalitions existed prior to the HCS or convened for primarily for the HCS) and other contextual factors thereby requiring some variation to the action planning process across and within states (table 1).

Action planning in each state is driven by coalitions, coalition workgroups and champions and is facilitated by a team comprising clinical and research faculty, core staff and community-based staff with expertise in community-engagement, data, OEND, MOUD and safer opioid

- Coalitions *must* select a minimum of five strategies for implementing evidence-based practices (EBPs) from the ORCCA menus including one involving each of the following
  - ORCCA Menu 1: Active OEND (i.e., proactively offer OEND to those at risk)
  - ORCCA Menu 2A: Expand MOUD access
  - ORCCA Menu 2B: Increase linkage to MOUD
  - ORCCA Menu 2C: Improve retention in MOUD
  - ORCCA Menu 3: Safer opioid prescribing/dispensing

- Priority sectors: Selected EBP strategies *must* include at least one to be implemented in each of the following:
  - Healthcare (i.e., outpatient healthcare centers, pre-hospital providers, emergency departments and urgent care, hospitals, primary care settings, public health departments, and pharmacies)
  - Behavioral health (i.e., substance use disorder and mental health treatment centers and social service agencies, including syringe service programs)
  - Criminal justice (i.e., pre-trial, jails, probation, parole, drug and problem-solving courts, police and "narcotics" task forces, halfway houses, community-based correctional facilities, and department of youth services)

- Key populations: Coalitions are encouraged to consider EBP strategies focused on those most vulnerable to opioid overdose:
  - People with a prior opioid overdose
  - People with reduced opioid tolerance
  - People who use other substances
  - People who have concomitant major mental illness
  - People who have a concomitant major medical illness
  - People who inject drugs

- Priority venues: Coalitions are encouraged to consider EBP strategies to be implemented in priority settings:
  - OEND: criminal justice settings, syringe service programs, emergency departments and acute care hospitals, sites of overdose for naloxone leave-behind, and mental health/addiction treatment programs
  - MOUD: criminal justice settings, primary care, general medical and behavioral/mental health settings, specialty addiction/substance use disorder treatment settings, and recovery programs
  - Safer opioid prescribing and dispensing: inpatient service, emergency/urgent care, outpatient clinics, ambulatory surgery, dental clinics, pharmacies

**Figure 1** Guidelines for HEALing Communities Study Community Action Planning Process to reduce opioid overdose deaths through overdose education and naloxone distribution, expansion of medications for opioid use disorder and safer opioid prescribing and dispensin. MOUD, medications for opioid use disorder; OEND, overdose education and naloxone distribution; ORCCA, Opioid-Overdose Reduction Continuum of Care Approach.

prescribing and dispensing. Coalition composition is described briefly in *Participant Involvement* and in detail elsewhere.[17] Of note, only in MA are *all* agencies to eventually be involved in EBP strategy implementation represented on the coalition. In other states, the agencies to be involved in implementation are represented to varying degrees across coalitions. In NY, community government partners (ie, health commissioners, county executives) are also involved in the action planning process.

### Step 1. Develop data-driven community goals

All coalitions begin the goal-setting process by reviewing community profiles and data dashboards, supplemented in one state (NY) by results of agent-based and systems dynamic modelling. The community profiles, developed through the CTH intervention, included lists of organisations in the community relevant to potential implementation of EBP strategies (eg, substance use disorder treatment centres, harm reduction programmes, emergency response agencies, hospitals). The data dashboards, also developed through the CTH intervention, contain data visualisations on community-specific trends in fatal and non-fatal opioid overdose, MOUD availability, naloxone distribution and high-risk opioid prescribing.[9]

During and following data review, coalitions discuss strengths, services, gaps and needs. The coalitions established goals varying in specificity across states and

**1) Develop data-driven community goals**
Coalitions review (1) lists of community organizations relevant to potential implementation of EBP strategies (e.g., treatment centers, harm reduction programs, emergency response agencies, hospitals), and (2) information about OEND, MOUD, and related services available through administrative data, local and contextualized data, and/or through phone surveys with agencies. Coalitions use goal setting tools, data visualization aids, and dashboards to describe strengths, services, gaps, and set goals.

**2) Discuss and prioritize EBP strategies**
Coalitions review decision aid tools (Supplementary Appendix) listing EBPs. Coalitions discuss relevant existing services, size of the gap between existing services and need, feasibility, and potential impact on opioid-involved overdose deaths. In some cases, coalitions score the EBPs on Likert scales and/or rank them according to impact and feasibility.

**3) Select EBP strategies to include in the community action plan**
Coalitions review narrative notes, scores, and rankings from the process described above and select by consensus or majority vote at least five EBPs, including one each from the following categories: active OEND, MOUD expansion, linkage to MOUD, retention in MOUD, and safer opioid prescribing and dispensing.

**4) Identify next steps for implementation of selected EBP strategies**
Coalitions suggest agencies that could partner to implement the selected strategies. Iteratively with coalition review, subject matter experts and staff review strategies to determine next steps for team members (e.g., initial meetings with partner organizations, contracting considerations, and technical assistance).

**Figure 2** Core components of the HEALing Communities Study Community Action Planning Process to reduce opioid overdose deaths through overdose education and naloxone distribution, medications for opioid use disorder and safer opioid prescribing and dispensing. EBP, evidence-based practice; MOUD, medications for opioid use disorder; OEND, overdose education and naloxone distribution; ORCCA, Opioid-Overdose Reduction Continuum of Care Approach.

communities (eg, 'Increase OEND', 'Two emergency departments will initiate MOUD by January 2021'). Coalitions set goals at the ORCCA menu level (ie, for OEND, MOUD and safer opioid prescribing/dispensing) or submenu level (ie, for active and passive OEND; for MOUD expansion, linkage and retention; for safer opioid prescribing/dispensing and disposal). States vary in the other factors their coalitions are asked to consider when setting goals; these factors include sustainability of EBP strategies (NY, MA), EBP strategies' reach to high-risk populations (OH, MA), ability to ensure health equity across ethnic and racial groups (MA, NY), and priority venues for EBP strategy implementation (OH, MA).

## Step 2. Discuss and prioritise EBPs that align with community goals

Following goal setting, coalitions consider specific strategies for implementing the EBPs included on the ORCCA menu and determine which are of highest priority for local implementation. In three states (OH, MA and NY), coalitions brainstorm and free-list EBP strategies that align with the study goals, priority sectors and/or venues and high-risk populations. In KY, coalitions are provided with initial lists of EBP strategies to consider that are prepopulated from the ORCCA by the academic team. Coalitions in all states consider the EBP strategies in terms of the existing services, size of the gap between existing services and need, feasibility and potential impact on opioid-involved overdose deaths. To frame the discussion of feasibility more concretely, coalitions in two states (OH

and NY) are asked to consider whether the EBP strategy is a new service, scale-up of an existing service via a new method of service delivery, new target population or new feature to an existing service. In two states (MA and NY), coalitions also consider sustainability. In all states, deliberations on the EBP strategies occur in full coalition meetings or in workgroups led by champions depending on the coalition's preference.

All coalitions begin by providing narrative feedback on each EBP strategy. One state's (KY) coalitions are asked to supplement the narrative feedback by scoring each EBP strategy in terms of size of the existing service gap, feasibility of implementation within the next 12 months and potential impact on opioid-involved overdose deaths using Likert scales. The scale scores are used to produce a summative priority score to help guide EBP strategy selection. Two states (NY and OH) ask coalitions to rank EBP strategies in terms of impact and feasibility, with the most impactful and feasible EBP strategies scoring highest. In one state (MA), coalitions prioritise EBP strategies through consensus building group dialogues as opposed to ranking.

In three states, coalitions are either required (MA) or encouraged to consider the cost of EBP strategy implementation (NY, OH). In MA, coalitions are provided with an overall budget cap to work within and as part of their planning are required to consider the cost of implementation as they make their EBP strategy selections. In NY, where communities receive funding for staffing and

**Table 1** Characteristics of action planning process that were unique to certain states

| Action planning steps | Characteristics | KY | MA | NY | OH |
|---|---|---|---|---|---|
| Step 1. Develop data-driven community goals | Supplementary factors considered by coalitions: | | | | |
| | ▶ Agent-based and systems dynamic modelling | | | X | |
| | ▶ EBP strategy sustainability | | X | X | |
| | ▶ Reach to high-risk populations and priority venues | | X | | X |
| | ▶ Equity across ethnic and racial groups | | X | X | |
| Step 2. Discuss and prioritise EBP strategies | Development of EBP strategy list considered by coalition | Pre-populated by the academic team | Generated by coalitions* | Generated by coalitions* | Generated by coalitions* |
| | Method of prioritising strategies | Likert scale ratings of size of current service gap, feasibility, and potential impact. | Consensus building group dialogues | Rank by impact and feasibility | Rank by impact and feasibility |
| | Coalition consideration of cost | Encouraged *not* to consider cost, but to focus on need, impact, and feasibility of each EBP strategy. | Provided with budget cap and required to consider costs | Encouraged, but not required | Encouraged, but not required |
| | Supplementary factors considered by coalitions: | | | | |
| | ▶ EBP strategy sustainability | | X | X | |
| | ▶ Method of EBP expansion (ie, new service, or scale-up of an existing service via a new method of delivery, target population, or feature) | | | X | X |
| Step 3. Select EBP strategies to include in action plan | Limits on strategy selections | 12 initial EBP strategies | No limit | No limit | No limit |
| Step 4. Identify next steps for implementation of selected EBP strategies | Coalition input on agencies to be involved in implementation | Coalitions <u>suggest</u> potential agencies to involve in implementation. | Coalition subgroups <u>determine</u> which agencies will be involved in implementation. | Coalitions <u>suggest</u> potential agencies to involve in implementation. | Coalitions <u>suggest</u> potential agencies to involve in implementation. |
| | Coalition input on next steps for beginning implementation | Coalition describes what they can do to support implementation and reach underserved populations who may experience disparities in access. | Coalitions estimate cost of each strategy with technical assistance from the academic team. | None beyond action plan established in Step 3 | None beyond action plan established in Step 3 |

*'Coalitions' is inclusive of full coalitions, coalition workgroups, and coalition champions with expertise in specific EBPs.
EBP, evidence-based practices; KY, Kentucky; MA, Massachusetts; NY, New York; OH, Ohio.

administrative costs and EBP strategy implementation based on population size, coalitions are encouraged, but not required, to consider the cost of EBP strategy implementation as they make their selections. In OH, all communities are provided with fixed and equivalent budgets with limits on expenditures for naloxone and technical assistance; coalitions consider these budget caps as they choose EBP strategies but are not asked to estimate the cost of individual EBP strategies. In KY, coalitions are encouraged *not* to consider cost, but rather to focus on potential need, impact, and feasibility of each EBP strategy.

## Step 3. Select EBP strategies to include in the community action plan

In full coalition and coalition subgroup meetings, members review narrative notes, scores and rankings from the process described above. In two states (KY and OH), coalitions are also provided with feedback from HCS faculty and staff on implementation feasibility with respect to structural or regulatory barriers. Coalitions weigh these considerations in the context of the guidelines described in figure 1 and choose through consensus or majority vote the EBP strategies to be implemented with support of HCS. In making their initial selections, coalition members understand that they can later add and refine EBP strategies as capacity expands. Only one state (KY) set a maximum for the number of EBP strategies that could be selected initially, not including OEND strategies already fast-tracked. Coalitions in KY are limited to 12 initial EBP strategies, requiring that they choose no more than three involving OEND, two for MOUD expansion, two for linkage to MOUD, two to improve MOUD retention and three focused on safer opioid prescribing/dispensing.

## Step 4. Identification of next steps for implementation of selected EBP strategies

Once EBP strategies are chosen, coalitions are encouraged to list partner agencies. In three states, coalition-listed agencies are suggested as potential partners for implementation (KY, NY, OH). In MA, the partner agencies for implementation are determined by coalition subgroups as part of the planning process and voted on by the coalitions, with the acceptation of agencies delivering office-based addictions treatment; these agencies were preselected by the study team. Coalition review of selected EBP strategies in one state (KY) also involves members describing what the coalition could do to support rollout (eg, write op-eds in the local paper in support of the EBP strategy, provide warm handoffs to agencies, etc) and ways to reach underserved populations with a primary focus on people of colour and Spanish-speaking individuals.

Iteratively with coalition review, HCS faculty and community staff in each state conduct internal reviews. In MA, state government partners are also involved in the internal review. Internal reviews focus on ensuring the EBP strategies selected adhere to the ORCAA menu and include the required high-risk populations and sectors, potential impact and feasibility. Internal teams also determine next steps for team members, such as initial meetings with partner organisations, contracting considerations and provision of technical assistance. In one state (MA), the review also includes planning for the mitigation of inequities associated with each EBP strategy and examination of each EBP strategy's cost as estimated by coalitions with technical assistance from community staff and HCS faculty.

On completion of the steps described above, descriptions of selected EBP strategies are provided to faculty and staff teams with expertise in implementation science, OEND, MOUD, safer opioid prescribing/dispensing and/or community engagement to start the process of working with agency partners and explore interest in and capacity for implementation of the coalitions' selected EBP strategies. This process varies from site to site and involves the development of an agency-specific implementation plan.

## Process and outcome assessment

Community research staff complete fidelity checklists monthly to document the extent to which all steps are followed and how long each step takes. EBP strategies selected by the coalitions are coded in the form of triads and entered into a harmonised data system shared across the four states. Specifically, the data are coded by venue type (eg, addiction treatment facility, social services agency, etc), sector (ie, criminal justice, behavioural health or healthcare) and ORCCA menu (ie, OEND, MOUD or safer opioid prescribing and dispensing). For each entry in the data set, additional data are recorded and updated throughout the study; these data include but are not limited to the date that the strategy was selected, when it was implemented, and whether it was successfully implemented. Analysis of the triad data is the focus of another paper and will involve enumeration of strategies selected across communities and by community characteristics. Data are also being collected on the strategies' reach (eg, number of naloxone units distributed, number of clients initiated on buprenorphine or involved in care navigation, etc). Cost of the intervention, including the community engagement process, is also being assessed, as described in detail elsewhere.[67] To collect data on coalition members' perspectives on the community engagement process, the HEALing Community Study includes repeated coalition member surveys and qualitative interviews which among many other domains, probes respondents on their experiences with the coalition and strengths and weaknesses of the partnership.[68]

## ETHICS AND DISSEMINATION

The HCS protocol was approved by a central Institutional Review Board (IRB; Advarra, Pro00038088) and is being conducted in accordance with The Code of Ethics of the World Medical Association (Declaration of Helsinki). As described elsewhere,[16] because no one person or group of people possess(es) the authority to give consent on behalf of all community members, investigators sought expert consultation and applied guidelines from the Ottawa Statement.[69] Because the CTH intervention poses no more than minimal risk to community members and the research could not be carried out otherwise (see 45.CFR.46.116), a waiver of informed consent was obtained for all community members who may be affected by the CTH intervention.

The action planning tools are disseminated to coalition members in hardcopy, via email and/or through a password-protected, online portal. We plan to disseminate updated versions of these tools to communities currently in the waitlist-control group via the same mechanisms when intervention begins in their communities. Results of the action planning process (ie, EBP strategy selections) are shared with community members, community agencies and community advisory boards through meetings, online and print media and presentations to community stakeholder groups in the intervention communities. In the future, we also plan to disseminate our findings at relevant conferences, meetings and through peer-reviewed journals. Finally, we will disseminate our findings, manuals, toolkits and other resources through federal partners including the National Institutes of Health, Substance Abuse and Mental Health Services Administration and the Centers for Disease Control and Prevention.

**Author affiliations**
[1]College of Public Health, University of Kentucky, Lexington, Kentucky, USA
[2]Department of Psychological Sciences, Purdue University, West Lafayette, Indiana, USA
[3]School of Social Work, Columbia University, New York, New York, USA
[4]School of Social Work, Boston University, Boston, Massachusetts, USA
[5]National Institute on Drug Abuse, National Institutes of Health, Bethesda, Maryland, USA
[6]Center for Applied Public Health Research, Research Triangle Institute, Research Triangle Park, North Carolina, USA
[7]Department of Psychiatry and Behavioral Neuroscience, University of Cincinnati College of Medicine, Cincinnati, Ohio, USA
[8]Center for Addiction Research, University of Cincinnati College of Medicine, Cincinnati, Ohio, USA
[9]General Internal Medicine-CARE Unit, Boston Medical Center, Boston, Massachusetts, USA
[10]Evans Department of Medicine, Boston Medical Center and Boston University School of Medicine, Boston, MA, USA
[11]Montefiore Hudson Valley Collaborative, Albert Einstein College of Medicine, Bronx, New York, USA
[12]College of Nursing, University of Kentucky, Lexington, Kentucky, USA
[13]College of Medicine, University of Kentucky, Lexington, Kentucky, USA
[14]College of Pharmacy, University of Kentucky, Lexington, Kentucky, USA
[15]Community & Workplace Health, Research Triangle International, Research Triangle Park, North Carolina, USA
[16]Clark County Health Department, Winchester, Kentucky, USA
[17]Franklin County Agency for Substance Abuse Policy Board, Frankfort, Kentucky, USA
[18]Department of Emergency Medicine, Ohio State University Wexner Medical Center, Columbus, Ohio, USA
[19]Center for Behavioral Health, Holyoke Medical Center, Holyoke, Massachusetts, USA
[20]Epiemiology and Population Health, Albert Einstein College of Medicine, Bronx, New York, USA

**Acknowledgements** We wish to acknowledge the participation of the HEALing Communities Study communities, community coalitions and Community Advisory Boards and state government officials who partnered with us on this study. We extend our sincere gratitude to the staff and community members who have dedicated their time and passion to this project and without whom the HEALing Communities Study would not be possible.

**Contributors** The following individuals contributed to the design of the action planning protocol described in this manuscript: AMY, PF, LSMG, LF, ML, JLB, MSL, TH, AB, BDR, RB-S, and TAB. The following individuals made substantial contributions to the implementation of the action planning protocol: AMY, PF, AF-B, LF, JG, CK, ML, HLS, SLW, JLB, MSL, TH, AB and MQ. The following individuals made substantial contributions to the drafted work and/or substantively revised it: AMY, JLB, TH, LSMG, RC, EO, TW, TAB, TB, RB-S, AB, AF-B, LF, LMG, PF, JG, CK, MSL, ML, MQ, BDR, HLS, SLW. All authors read and approved the final manuscript and agree to be accountable for all aspects of the work.

**Funding** This research was supported by the National Institutes of Health through the NIH HEAL (Helping to End Addiction Long-term[SM]) Initiative under award numbers UM1DA049394, UM1DA049406, UM1DA049412, UM1DA049415, UM1DA049417 (ClinicalTrials.gov Identifier: NCT04111939). The Substance Abuse and Mental Health Services Administration (SAMHSA) is a non-funding sponsor of the study. This study protocol (Pro00038088) was approved by Advarra Inc., the HEALing Communities Study single Institutional Review Board. We wish to acknowledge the participation of the HEALing Communities Study communities, community coalitions, and Community Advisory Boards and state government officials who partnered with us on this study. The content is solely the responsibility of the authors and does not necessarily represent the official views of the National Institutes of Health, the Substance Abuse and Mental Health Services Administration or the NIH HEAL Initiative[SM]. R. Chandler was substantially involved in this paper, consistent with her role as Scientific Officer.

**Competing interests** Competing interest disclosures include the following: LSM is a consultant for the Boston Public Health Commission, Action for Boston Area development and the Young Foundation, SLW has served as a scientific advisor to Opiant Pharmaceuticals, Titan Pharmaceuticals, Astra Zeneca, Cerevel, Arbor Pharmaceuticals, Brainsway, Summit Biosciences, Trevi and Camurus. JLB ad ML receive investigator-initiated support paid to the institution from Gilead Sciences, Inc. MRL has consulted for Titan Pharmaceuticals and Camurus. AF-B is the Program Director of Voices of Hope, a University of Kentucky HCS partner organisation. PF, HLS, AMY, EO, LMG, RC, JG, CK and LF have no competing interests.

**Patient and public involvement** Patients and/or the public were involved in the design, or conduct, or reporting, or dissemination plans of this research. Refer to the Methods section for further details.

**Patient consent for publication** Not applicable.

**Provenance and peer review** Not commissioned; externally peer reviewed.

**ORCID iD**
April M Young http://orcid.org/0000-0003-3969-3249

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
