## [Reviewer comments · BMJ Open]

ARTICLE DETAILS

TITLE (PROVISIONAL)	Protocol for community-driven selection of strategies to implement evidence-based practices to reduce opioid overdoses in the HEALing Communities Study, a trial to evaluate a community-engaged intervention in Kentucky, Massachusetts, New York, and Ohio
AUTHORS	Young, April; Brown, Jennifer; Hunt, Timothy; Sprague Martinez, Linda; Chandler, Redonna; Oga, Emmanuel; Winhusen, Theresa; Baker, Trevor; Battaglia, Tracy; Bowers-Sword, Rachel; Button, Amy; Fallin-Bennett, Amanda; Fanucchi, Laura; Freeman, Patricia; Glasgow, LaShawn; Gulley, Jennifer; Kendell, Charles; Lofwall, Michelle; Lyons, Michael S.; Quinn, Maria; Rapkin, Bruce; Surratt, Hilary; Walsh, Sharon

VERSION 1 – REVIEW

REVIEWER	Ferri, Marica European Monitoring Centre for Drugs and Drug Addiction, Public Health Unit
REVIEW RETURNED	15-Feb-2022

GENERAL COMMENTS	Thank you for this interesting, original and timely protocol of research: I think this is worth publication, and it needs to be clearer on some points: 1. Naloxone is a life-saving intervention, and it is not clear to me if you are providing it to all the clusters you will compare. If you are not, I see an important ethical issue that you need to revise.2. In the list of evidence-based interventions, you also list extended-release Naltrexone, an opioid antagonist without the same objectives and the same evidence in support as supervised injected medical heroin and drugs consumption facilities that you do not include in your protocol.3. You use an appropriate C-RCT, but you do not discuss the type of analysis and how you will avoid the possible bias (I suggest you check your methods against these guidelines: https://pubmed.ncbi.nlm.nih.gov/22951546/). My last point is about the outcomes. Although I appreciate the process through which the community select their outcomes, do not forget 2 points: provide the communities with guidance on the implementation and include some outcomes (reduction of mortality?) equal for all to make sure you can compare. I hope these comments are helpful, and I will be happy to review your paper again.
---

REVIEWER	Kreiner, Peter Brandeis University
-----------------	---------------------------------------

REVIEW RETURNED	18-Feb-2022
-------------

GENERAL COMMENTS	This is a study protocol for one phase of the Communities that Heal (CTH) project, namely participant community selection of strategies for implementing evidence-based programs to reduce opioid-involved overdose deaths. The paper provides a thorough and highly detailed description of the steps communities are to undertake as part of this process, as well as description of the variations in aspects of the process across four states. The description is clear and concise. What seems to me to be missing, however, is any discussion of how this phase is to be assessed as part of the CTH study. Evidently communities will each generate a list of strategies. How is the appropriateness of these strategies to community context and local vulnerable populations to be assessed? Is assessment of appropriateness based solely on whether all the steps of this phase were followed and documented? As noted in the text, an important feature of CTH is enabling communities to apply local knowledge to their needs. As noted in the strengths and limitations, facilitation of the action planning process is key to allowing all voices in the community to be heard. Will this facilitation be assessed in some way? Without some discussion of data to be collected and how it will be used to assess accomplishment of this phase, I don't see how this protocol gives rise to a study.
--

VERSION 1 – AUTHOR RESPONSE

Reviewer: 1

Dr. Marica Ferri, European Monitoring Centre for Drugs and Drug Addiction

Comments to the Author:

Dear author,

Thank you for this interesting, original and timely protocol of research: I think this is worth publication, and it needs to be clearer on some points:

1. Naloxone is a life-saving intervention, and it is not clear to me if you are providing it to all the clusters you will compare. If you are not, I see an important ethical issue that you need to revise.

We appreciate the reviewer's comment and wholeheartedly agree. Overdose education and naloxone distribution (OEND) is a required component of the study, meaning that communities involved in the study cannot opt out of OEND. They must choose an evidence-based practice that expands OEND. We address this in the Community Action Planning Guidelines section, "Coalitions are required to select a minimum of five strategies, including at least one to implement each required EBP from the ORCCA: (a) expansion of proactive offering of OEND to at risk individuals and members of their social networks, (b) expansion of buprenorphine and/or methadone access, (c) improved linkage to MOUD, (d) increased retention in MOUD, and (e) promotion of safer opioid prescribing/dispensing." We have underlined "required" in our revision to add emphasis and make sure that it is brought to the reader's attention. We also want to note that Wave 1 and Wave 2 communities' naloxone programs that were in place prior to the study are not being disrupted - rather the study intervention is enhancing what the communities already have in place.

2. In the list of evidence-based interventions, you also list extended-release Naltrexone, an opioid antagonist without the same objectives and the same evidence in support as supervised injected medical heroin and drugs consumption facilities that you do not include in your protocol.

We agree with the reviewer's comments as they related to extended-release naltrexone. While we discuss with coalitions extended-release naltrexone as an FDA-approved medication for opioid use disorder, communities were required to choose expansion with methadone and/or buprenorphine because these have the strongest evidence for mortality reduction. Coalitions could also choose naltrexone, but they could not choose to expand treatment with naltrexone only. We address this in item b in the sentence included in the Community Action Planning Guidelines section, "Coalitions are required to select a minimum of five strategies, including at least one to implement each required EBP from the ORCCA: (a) expansion of proactive offering of OEND to at risk individuals and members of their social networks, (b) expansion of buprenorphine and/or methadone access, (c) improved linkage to MOUD, (d) increased retention in MOUD, and (e) promotion of safer opioid prescribing/dispensing."

The reviewer is correct in noting that the "menu" of options from which coalitions could choose does not include supervised injected medical heroin and drugs consumption facilities. While we agree that these interventions are very promising in having the potential to reduce overdose deaths and that more research is needed on these interventions, we could not include them in the menu because evidence on their implementation in the U.S. is currently limited and because state policies in the four states involved did not allow for the establishment of such facilities at the onset of the study.

3. You use an appropriate C-RCT, but you do not discuss the type of analysis and how you will avoid the possible bias (I suggest you check your methods against these guidelines:

<https://nam04.safelinks.protection.outlook.com/?url=https%3A%2F%2Fpubmed.ncbi.nlm.nih.gov%2F22951546%2F&data=05%7C01%7Ccapril.young%40uky.edu%7Cdc1de0e0a20b43207f0508da375e20b7%7C2b30530b69b64457b818481cb53d42ae%7C0%7C0%7C637883174655989598%7CUnknown%7CTWFpbGZsb3d8eyJWljoImC4wLjAwMDAiLCJQIjoiV2luMzliLCJBTiI6IjEhaWwiLCJXVCi6Mn0%3D%7C3000%7C%7C&sdata=FuRWjVS4As8WSS9KbkBvLHyTd2y8LSwXskopBYcTIAI%3D&reserved=0>).

The reviewer raises a great point. The analysis of a trial of this magnitude and this design can be complex. Because this paper focuses on the protocol for communities' action planning process, we do not delve into the statistical approach for analysis of epidemiological and administrative data obtained for evaluation of the overall trial. However, the consortium recently published an entire paper dedicated to describing the analysis of this trial (see below). We have added a citation to the paper so that readers can easily find more information if they are interested.

Westgate PM, Cheng DM, Feaster DJ, Fernández S, Shoben AB, Vandergrift N. Marginal modeling in community randomized trials with rare events: Utilization of the negative binomial regression model. *Clin Trials*. 2022 Apr;19(2):162-171. doi: 10.1177/17407745211063479. Epub 2022 Jan 6. PMID: 34991359; PMCID: PMC9038610.

My last point is about the outcomes. Although I appreciate the process through which the community select their outcomes, do not forget 2 points: provide the communities with guidance on the implementation and include some outcomes (reduction of mortality?) equal for all to make sure you can compare.

We appreciate this point. All communities are striving toward the same primary and secondary outcomes. As mentioned in the response to the Editor's comment, we have added information on the primary outcome and reference to a paper that describes the outcomes in detail for this trial: Slavova S, LaRochelle MR, Root ED, Feaster DJ, Villani J, Knott CE, Talbert J, Mack A, Crane D, Bernson D, Booth A, Walsh SL. Operationalizing and selecting outcome measures for the HEALing Communities

Study. Drug Alcohol Depend. 2020 Dec 1;217:108328. doi: 10.1016/j.drugalcdep.2020.108328. Epub 2020 Oct 2. PMID: 33091844; PMCID: PMC7531340. Also, although the communities have some flexibility in their action planning process, there are guidelines (Figure 1) and core components (Figure 2) common to all sites that communities must follow to allow for comparison. The community coalitions are not responsible for the actual implementation of evidence-based practices; research site staff with expertise in implementation science, medication for opioid use disorder, naloxone distribution, and safe opioid prescribing and disposal work with local partner agencies to implement the evidence-based practices and provide technical assistance.

I hope these comments are helpful, and I will be happy to review your paper again.

Reviewer: 2

Dr. Peter Kreiner, Brandeis University

Comments to the Author:

This is a study protocol for one phase of the Communities that Heal (CTH) project, namely participant community selection of strategies for implementing evidence-based programs to reduce opioid-involved overdose deaths. The paper provides a thorough and highly detailed description of the steps communities are to undertake as part of this process, as well as description of the variations in aspects of the process across four states. The description is clear and concise.

We appreciate the reviewer's positive feedback.

What seems to me to be missing, however, is any discussion of how this phase is to be assessed as part of the CTH study. Evidently communities will each generate a list of strategies. How is the appropriateness of these strategies to community context and local vulnerable populations to be assessed? Is assessment of appropriateness based solely on whether all the steps of this phase were followed and documented?

The reviewer raises an important question. Fidelity to the process is assessed on the basis of whether all steps of the phase were followed and documented and whether key milestones were achieved. The assessment of the strategy selection is multi-faceted. Once strategies are selected, the strategy is coded into a triad and entered into a harmonized data system shared across the four states. Specifically, the data are coded by venue type (e.g., addiction treatment facility, social services agency, etc), sector (i.e., criminal justice, behavioral health, or healthcare), and evidence-based practice menu known as the ORCCA (i.e., OEND, MOUD, or safer opioid prescribing and dispensing). For each entry in the dataset, additional data are recorded and updated throughout the study; these data include but are not limited to the date that the strategy was selected, when it was implemented, and whether it was successfully implemented. Analysis of the triad data will involve enumeration of strategies selected across communities and by community characteristics. Data are also being collected on the strategies' reach (e.g., number of naloxone units distributed, number of clients involved in care navigation, etc). Cost of the intervention, including the community engagement process, is also being assessed, as described in the first citation below. To collect data on coalition members' perspectives on the community engagement process, the HEALing Community Study includes repeated coalition member surveys and qualitative interviews which among many other domains, probes respondents on their experiences with the coalition and strengths and weaknesses of the partnership. This data collection is described in detail in the second citation below. We have added this information and the citations to the manuscript.

Aldridge AP, Barbosa C, Barocas JA, Bush JL, Chhatwal J, Harlow KJ, Hyder A, Linas BP, McCollister KE, Morgan JR, Murphy SM, Savitzky C, Schackman BR, Seiber EE, E Starbird L, Villani J, Zarkin GA. Health economic design for cost, cost-effectiveness and simulation analyses in the HEALing Communities Study. *Drug Alcohol Depend.* 2020 Dec 1;217:108336. doi: 10.1016/j.drugalcdep.2020.108336. Epub 2020 Oct 3. PMID: 33152672; PMCID: PMC7532345. Knudsen HK, Drainoni ML, Gilbert L, Huerta TR, Oser CB, Aldrich AM, Campbell ANC, Crable EL, Garner BR, Glasgow LM, Goddard-Eckrich D, Marks KR, McAlearney AS, Oga EA, Scalise AL, Walker DM. Model and approach for assessing implementation context and fidelity in the HEALing Communities Study. *Drug Alcohol Depend.* 2020 Dec 1;217:108330. doi: 10.1016/j.drugalcdep.2020.108330. Epub 2020 Oct 2.

As noted in the text, an important feature of CTH is enabling communities to apply local knowledge to their needs. As noted in the strengths and limitations, facilitation of the action planning process is key to allowing all voices in the community to be heard. Will this facilitation be assessed in some way?

We agree that assessment of the facilitation is important and that these details are important to include in the paper. The HEALing Communities Study includes longitudinal coalition member surveys and qualitative interviews which among many other domains, probes respondents on their experiences with the coalition, including strengths and weaknesses of the partnership. This part of the HEALing Communities Study protocol is described in detail elsewhere: Knudsen HK, Drainoni ML, Gilbert L, Huerta TR, Oser CB, Aldrich AM, Campbell ANC, Crable EL, Garner BR, Glasgow LM, Goddard-Eckrich D, Marks KR, McAlearney AS, Oga EA, Scalise AL, Walker DM. Model and approach for assessing implementation context and fidelity in the HEALing Communities Study. *Drug Alcohol Depend.* 2020 Dec 1;217:108330. doi: 10.1016/j.drugalcdep.2020.108330. Epub 2020 Oct 2. We have added information and a citation to this paper in our revised manuscript. Without some discussion of data to be collected and how it will be used to assess accomplishment of this phase, I don't see how this protocol gives rise to a study.

We agree with the reviewer that our omission of these details resulted in an understatement of the rigor of this process and its eventual evaluation. With the addition of the information described above, we hope that the protocol in the context of a study is more clear.

Reviewer: 1
Competing interests of Reviewer: None

Reviewer: 2
Competing interests of Reviewer: None

Thank you for your continued consideration of our manuscript,

April Young
Associate Professor
University of Kentucky

VERSION 2 – REVIEW

REVIEWER	Ferri, Marica
-----------------	---------------

	European Monitoring Centre for Drugs and Drug Addiction, Public Health Unit
REVIEW RETURNED	25-Jun-2022

GENERAL COMMENTS	Thank you for this important piece of work on the Community Coalition's implementation of strategies to reduce opioid overdoses. The topic is extremely urgent, and grants for speed publication and community-based approaches are promising approaches to implementation. Nevertheless, your manuscript has important issues to correct to help the reader follow your methods. In the first place, reading your title, I expected to find results of a multisite parallel-group cluster randomised wait-list controlled trial. I had to read it all twice to understand that you were only discussing a selection of strategies for implementation in the present article. I am sure you can edit your title to clarify the expectations. Secondly, in the methods section, you should decide whether you describe your cluster-randomised study (including power calculation, intra and inter-cluster calculations (http://www.consort-statement.org/extensions?ContentWidgetId=554); or you describe the methods applied by the communities to select the strategies (what you call the "facilitation by decision tools"). In addition, I believe it would be clearer if you distinguished between the strategy, the methods to choose those strategies AND the outcomes expected from implementing the procedures. You can probably add a flowchart and graphic representations of the process. I suspect that the audience of this publication should be decision-makers and local NGOs that can benefit from an easily implementable implementation process. Beyond the scope of this publication, I wonder whether you are thinking of preparing some interactive e-facilitated decision and implementation tools. I would be more than happy to be further informed about possible adoption in a European context. Thank you.
--

VERSION 2 – AUTHOR RESPONSE

Reviewer 1

Thank you for this important piece of work on the Community Coalition's implementation of strategies to reduce opioid overdoses. The topic is extremely urgent, and grants for speed publication and community-based approaches are promising approaches to implementation. Nevertheless, your manuscript has important issues to correct to help the reader follow your methods.

*We appreciate the reviewer's positive feedback and recognition of the importance and urgency of disseminating information on the protocol.

In the first place, reading your title, I expected to find results of a multisite parallel-group cluster randomised wait-list controlled trial. I had to read it all twice to understand that you were only discussing a selection of strategies for implementation in the present article. I am sure you can edit your title to clarify the expectations.

*The reviewer raises a valid point. The instructions for authors requests that manuscript titles include the details on the study design. As such, we included information on the design of the overall HEALing Communities Study in the title. Then, in first round of review of our manuscript, the Editor requested, "Please revise the title of your manuscript to include the research question, study design, and setting. This is the preferred format of the journal." In response, we expanded the title to include more details about the HEALing Communities Study design. However, we agree with the reviewer that describing the study on the whole rather than the specific protocol component that is described in the manuscript creates confusion. In response, we have now re-revised our title to read, "Community-driven selection of strategies to implement evidence-based practices to reduce opioid overdoses in the HEALing Communities Study, a trial to evaluate a community-engaged intervention in Kentucky, Massachusetts, New York, and Ohio."

Secondly, in the methods section, you should decide whether you describe your cluster-randomised study (including power calculation, intra and inter-cluster calculations; or you describe the methods applied by the communities to select the strategies (what you call the "facilitation by decision tools").

*We appreciate this comment and expect that it stemmed from the extraneous detail included in the manuscript title. The focus of this manuscript is on the latter (i.e., community selection of strategies), the former has already been described in great detail elsewhere (citation below). We believe that the revisions made to the title in response to the comment above will help to clarify the scope of the paper.

Westgate PM, Cheng DM, Feaster DJ, Fernández S, Shoben AB, Vandergrift N. Marginal modeling in community randomized trials with rare events: Utilization of the negative binomial regression model. *Clin Trials*. 2022 Apr;19(2):162-171. doi: 10.1177/17407745211063479. Epub 2022 Jan 6. PMID: 34991359; PMCID: PMC9038610.

In addition, I believe it would be clearer if you distinguished between the strategy, the methods to choose those strategies AND the outcomes expected from implementing the procedures.

*This is a good suggestion. The paper was largely organized in this order with subsections aligning with the three components noted by the reviewer, but the sections were labeled in a way that obscured that organization. To make the manuscript clearer, we have revised the section headings to explicitly indicate where strategies, the selection process, and outcome assessment are described.

You can probably add a flowchart and graphic representations of the process.

*We appreciate the reviewer’s suggestion. Drawing on information that was presented tabularly in Figure 2 in the original manuscript, we have created a new Figure 2 which graphically represents the stepwise process. In addition, the supplementary appendix contains images of the tools used in the process in each of the states.

I suspect that the audience of this publication should be decision-makers and local NGOs that can benefit from an easily implementable implementation process.

*We agree with the reviewer that in addition to academic researchers designing community-engaged interventions, local NGOs and decision-makers could benefit from the information included in this publication. Our hope is that the inclusion of the documents and tools for each state in the Supplementary Appendix will be of use to those in the practice community. We also intend to manualize key components of the intervention at the conclusion of the HEALing Communities Study.

Beyond the scope of this publication, I wonder whether you are thinking of preparing some interactive e-facilitated decision and implementation tools.

*We agree with the reviewer that an e-facilitated process and interactive tools would be beneficial for communities who have access to the technology to guide them through the process. Research sites will draw on their experience implementing the action planning process in Wave 1 communities to improve the process for Wave 2. For example, at the Kentucky site, the research team is programming the action planning tools into an interactive community web portal and will be piloting those interactive versions of the tools with the trial’s Wave 2 community coalitions. Upon conclusion of the action planning process in Wave 2, the team’s hope is to disseminate information on coalitions’ experience with those platforms and potentially also the platforms so that they can be used by other communities beyond the study.

VERSION 3 – REVIEW

REVIEWER	Ferri, Marica European Monitoring Centre for Drugs and Drug Addiction, Public Health Unit
REVIEW RETURNED	11-Aug-2022

GENERAL COMMENTS	Dear authors, thank you for thoughtfully considering my suggestions. I liked the clarity of your explanations and the improvement of your manuscript. As an author and an editor, I appreciate the frustration that many and sometimes conflicting requests create. I appreciate your genuine effort to accommodate all of them to improve the readability of your publication. I think it is ready for publication and widespread dissemination in the US and beyond, wherever opioid overdoses are destroying lives and adding unnecessary burden to the health systems.
--

VERSION 3 – AUTHOR RESPONSE

Reviewer: 1

Dr. Marica Ferri, European Monitoring Centre for Drugs and Drug Addiction

Comments to the Author:

Dear authors, thank you for thoughtfully considering my suggestions. I liked the clarity of your explanations and the improvement of your manuscript. As an author and an editor, I appreciate the frustration that many and sometimes conflicting requests create. I appreciate your genuine effort to accommodate all of them to improve the readability of your publication. I think it is ready for publication and widespread dissemination in the US and beyond, wherever opioid overdoses are destroying lives and adding unnecessary burden to the health systems.

Reviewer: 1

Competing interests of Reviewer: No competing interest

We appreciate the reviewer's positive feedback and recognition of the importance of the project.